# Insertion of Transposable Elements in *AVR-Pib* of *Magnaporthe oryzae* Leading to LOSS of the Avirulent Function

**DOI:** 10.3390/ijms242115542

**Published:** 2023-10-24

**Authors:** Jinbin Li, Lin Lu, Chengyun Li, Qun Wang, Zhufeng Shi

**Affiliations:** 1 The Ministry of Agriculture and Rural Affairs International Joint Research Center for Agriculture, The Ministry of Agriculture and Rural Affairs Key Laboratory for Prevention and Control of Biological Invasions, Yunnan Key Laboratory of Green Prevention and Control of Agricultural Transboundary Pests, Agricultural Environment and Resource Research Institute, Yunnan Academy of Agricultural Sciences, Kunming 650205, China; qunwang70@163.com (Q.W.);; 2Flower Research Institute, Yunnan Academy of Agricultural Sciences, Kunming 650205, China; lu_lin2005@sina.com; 3The Ministry of Education Key Laboratory for Agricultural Biodiversity and Pest Management, Yunnan Agricultural University, Kunming 650200, China; licheng_yun@163.com

**Keywords:** *Magnaporthe oryzae*, rice blast, *AVR-Pib*, evolution, transposable element (TE)

## Abstract

Rice blast is a very serious disease caused by *Magnaporthe oryzae*, which threatens rice production and food supply throughout the world. The avirulence (*AVR*) genes of rice blast are perceived by the corresponding rice blast resistance (*R*) genes and prompt specific resistance. A mutation in *AVR* is a major force for new virulence. Exploring mutations in *AVR* among *M. oryzae* isolates from rice production fields could aid assessment of the efficacy and durability of *R* genes. We studied the probable molecular-evolutionary patterns of *AVR-Pib* alleles by assaying their DNA-sequence diversification and examining their avirulence to the corresponding *Pib* resistance gene under natural conditions in the extremely genetically diverse of rice resources of Yunnan, China. PCRs detected results from *M. oryzae* genomic DNA and revealed that 162 out of 366 isolates collected from Yunnan Province contained *AVR-Pib* alleles. Among them, 36.1–73.3% isolates from six different rice production areas of Yunnan contained *AVR-Pib* alleles. Furthermore, 36 (28.6%) out of 126 isolates had a transposable element (TE) insertion in *AVR-Pib*, which resulted in altered virulence. The TE insertion was identified in isolates from rice rather than from *Musa nana* Lour. Twelve *AVR-Pib* haplotypes encoding three novel *AVR-Pib* variants were identified among the remaining 90 isolates. *AVR-Pib* alleles evolved to virulent forms from avirulent forms by base substitution and TE insertion of Pot2 and Pot3 in the 5′ untranslated region of *AVR-Pib*. These findings support the hypothesis that functional *AVR-Pib* possesses varied sequence structures and can escape surveillance by hosts via multiple variation manners.

## 1. Introduction

In the coevolution of plants and pathogens, the latter can adapt to the host and environment, and selection is the major evolutionary force. Up to now, the “arms race” and “trench warfare” hypotheses of coevolution between host resistance (*R*) genes and pathogen avirulence (*AVR*) genes have been proposed. In the principal hypothesis of the arms-race, the mutation of *R* genes and *AVR* genes is derived by directional selection. Contrarily, it is derived by undirectional selection in the trench-warfare hypothesis [1]. 

Rice blast is one of the most serious diseases in rice worldwide, which caused by the fungus *Magnaporthe oryzae*. The application of rice varieties with multiple resistant genes is the most important method controlling the disease with an economical, environmentally and friendly manner. The resistance of the sole resistance gene rice variety to *M*. *oryzae* can be lost quickly by the high variation of fungus. Up to now, more than 35 rice blast *R* genes have been cloned in rice: *Pita*, *PiCO39*, *Pish*, *Pi1*, *Pik*, *Pikp*, *Pikh/Pi54*, *Pikm*, *Pb1*, *Pid3*, *Pia*, *Pib*, *Pid2*, *Pit*, *Pizt*, *Pi2*, *Pi5*, *Pi9*, *pi21*, *Pi25*, *Pi36*, *Pi37*, *Pi56*, *Pi63*, *Pi35*, *Pid3-4A*, *Pi50*, *Pii*, *Pi54*, *Pike*, *Piks* [2,3].

The effectiveness of resistance of *Pib* has been examined in different rice production provinces in China. *Pib* exerts a high level of resistance to *M. oryzae* from Heilongjiang Province, and can be applied as a parent for resistance breeding in Heilongjiang Province [4]. *Pib* is moderately resistant in Fujian Province [5], but *Pib* exhibits partial resistance in Guangdong, Sichuan and Guizhou Provinces [6,7]. Different resistance spectra of *Pib* were detected in 282 blast isolates collected from *Indica* rice- and *Japonica* rice production regions in Yunnan Province [8]. Those results showed that the *Pib* gene exhibits different resistance to the Chinese rice blast from different rice-growing regions. There were 42 out of 54 varieties containing *Pib* in Chinese elite hybrid rice varieties, two haplotypes of *Pib* were identified and 9 different haplotypes of *AVR-Pib* were found among 27 *M. oryzae* [9]. The reactions of differential isolates on 54 rice varieties between the frequency of *AVR-Pib* haplotypes in the differential isolates showed a good correlation [9]. They showed that the *Pib* gene was widely distributed in rice varieties in China, and the adapt variation of *AVR-Pib* of *M. oryzae* occurred. The rice resistance gene of *Pib* is located on the long arm of chromosome 2 [10,11]. The cDNA length of the *Pib* gene contains 306 bp of 5′ untranslated regions (UTRs), 3753 bp of open reading frames (ORFs) (containing 3 exons) and 229 bp of 3′ UTRs, and encodes a NBS-LRR protein with 1251 amino acids [12]. In China, 16 out of 204 (7.8%) varieties have been detected hoarding the *Pib* gene in a mini-core collection of Chinese rice germplasm using their functional markers [13]. Moreover, 11 varieties were identified with the *Pib* gene among 58 leading rice cultivars or hybrid rice parents in China using functional DNA markers [14]. *Pi-b* was detected in 33 landraces among 176 landraces (18.8%) from Yunnan Province in China [15]. *Pib* gene homologs (87-bp deletion in exon 1 of *Pib*, leading to a loss of the resistance function of *Pib*) were identified in Yunnan Yuanjiang type of common wild rice (*Oryza rufipogon* Griff) [16]. In the Philippines, 32 out of 52 commercial rice varieties have been shown to contain *Pib* as detected by polymerase chain reaction (PCR) primers specific to *Pib* [17]. They showed that the *Pib* gene was widely used in rice breeding in China.

Based on the gene-for-gene theory, rice *R* gene(s) can discern the corresponding *AVR* of *M. oryzae* and trigger the defense response to prevent invasion. So far, 12 *AVR* genes have been cloned in *M. oryzae*: *AVR-Pib* [18], *AVR-Pi54* [19], *AVR-Pi9* [20], *AVR-Pia* [21], *AVR-Pik/km/kp* [21], *AVR-Pii* [21], *AVR-Pizt* [22], *ACE1* [23], *AVR1-CO39* [24], *AVR-Pita* [25], *PWL1* [26], and *PWL2* [27]. The *AVR-Pib* allele of *M. oryzae* predicts the resistance efficacy of the rice *R* gene *Pib*. *AVR-Pib* encodes a putative secreted protein with 75 amino acids. *AVR-Pib* is perceived by the host as a Pib resistance protein and prompts the innate immune response [18]. Transposable element (TE) insertion, segmental deletion, absence and point mutations have been identified in *AVR-Pib* in 60 rice blast isolates from Guangdong, Hunan, Liaoning, Jilin and Heilongjiang of Provinces in China, which resulted in a loss of avirulence [18]. Only TE insertion has been observed in *AVR-Pib* in 248 *M. oryzae* isolates from the Philippines [17].

Further clarification of the diversification and evolution of the *AVR* gene is useful for the prediction of the effectiveness and durability of *R* genes for resistance breeding. Here, we wished to: (i) detect the diversity of nucleotide sequences of *AVR-Pib* alleles of *M. oryzae* under field conditions; (ii) determine the avirulence function of *AVR-Pib* variations to the *Pib* gene; (iii) reveal the molecular diversification principles of *AVR-Pib* alleles in *M. oryzae* in Yunnan Province. Our results provide useful information for rice-blast disease controlling and resistant breeding in China. 

## 2. Results

### 2.1. Effectiveness of the Pib Gene and Frequency of AVR-Pib Alleles

The efficacy of the *Pib* gene was examined by pathogenicity assays. A total of 223 of the 366 *M. oryzae* isolates tested were avirulent to the *Pib* gene-containing rice monogenic line IRBLb-B (Table 1). The percentage of avirulent isolates to *Pib* was 60.9%, whereas the remaining 143 isolates were virulent to the *Pib* gene (Table 1). The percentage of the avirulent isolate was 100, 75.9, 75.0, 57.6, 53.6, and 48.2% in northwestern, central, northeastern, southeastern, southwestern and western Yunnan Province, respectively. Among 366 isolates, 44.3% of isolates with the *AVR-Pib* allele were amplified by *AVR-Pib*-specific primers (AVR-Pib F1/AVR-Pib R1) (Table 1; Appendix A), and three genotypes (L1 with 1231 bp, L2 with 3100 bp and L3 with both 1231 bp and 3100 bp) of *AVR-Pib* alleles in 162 isolates were amplified (Table 1; Appendix A). The highest percentage of amplification of *AVR-Pib* was 73.3% in the rice blast isolates collected from northwestern Yunnan Province, whereas the lowest percentage was 36.1% from northeastern Yunnan Province (Table 1). The percentage of *AVR-Pib* was 46.3, 36.1, 73.3, 51.5, 67.9 and 39.0% in central, northeastern, northwestern, southeastern, southwestern and western Yunnan Province, respectively. The percentage of *AVR-Pib* was 47.0 and 42.4% in *Xian*/*Indica* (*XI*) rice- and *Geng*/*Japonica* (*GJ*) rice production areas in Yunnan. The genotype of L1, L2 and L3 alleles of *AVR-Pib* was detected in 104, 53 and 5 isolates, with percentages of 28.4%, 14.5% and 1.3%, respectively (Table 1). The genotype of L1, L2 and L3 alleles of *AVR-Pib* was detected in the *XI* rice production area, whereas L3 was absent in the *GJ* rice production area (Table 1).

### 2.2. Virulence Function of AVR-Pib Variations against the Pib Gene

Twelve *AVR-Pib* haplotypes (H01 to H12) (Table 2), excluding the original *AVR-Pib* allele (GenBank accession number, KM887844), were detected on the nucleotide sequence assemblies of 90 isolates of L1 alleles containing a 719-bp 5′-region, 225-bp coding DNA sequence (CDS) and 302-bp 3′-region of *AVR-Pib* (Table 2; Appendix A). Moreover, insertion of Pot2 (at position −275) and Pot3 (at position −240) was identified based on the DNA sequence assemblies of six and 30 isolates (Table 2; Figure 1), respectively, and the amplicon size difference between L1 and L2 (Appendix A). The 12 novel *AVR-Pib* haplotypes (H01–H12) were identified compared to previous published alleles [3,9]. Alignment of DNA sequence assemblies of the *AVR-Pib* allele from 90 isolates revealed 18 mutation sites, including six mutant sites in the CDS region which were not in the signal–peptide region (Table 2; Appendix A). Six mutant sites in the CDS region led to changes in amino acids (Table 3). The CDS sequence assemblies of the *AVR-Pib* allele among the 126 isolates (including L1, L2 and L3) were predicted to produce four AVR-Pib proteins (Table 3). Among them, amino acid variations were predicted to occur at six positions (Table 3). Amino-acid variations at F54L in H05 and H06, E46V, F53S and F54V in H07 were found; these isolates of the corresponding haplotypes were avirulent on the monogenic line IRBLb-B (with *Pib*). Meanwhile, the amino acid variations at F47L, I49T and R50G were found in one isolate with H08, which was virulent on the monogenic line IRBLb-B (with *Pib*) (Table 3) and the amino acid variations at F47, I49 and R50 in H01, H02, H03, H04, H05, H06, H07, H09, H10 and H12; these isolates were avirulent on the monogenic line IRBLb-B (with *Pib*), whereas the amino acid variations at 47L, 49T and 50G in H08, as well as the isolate, were virulent on IRBLB-b (Table 3). This finding suggested that the amino acids F47, I49 and R50 were crucial for the avirulence function of *AVR-Pib*. 

The different haplotypes of H01, H02, H03, H04, H09, H10, H11 and H12 had no change on amino acid sequence (Table 3). Three-dimensional protein structures built by homology modeling (SWISS-MODEL; https://swissmodel.expasy.org/, accessed on 18 February 2019) showed the different protein structures of these four (H1, H5, H7 and H8) AVR-Pib variants (Appendix A). Isolates of H01 (amino acids that were the same as that with a GenBank accession number of KM887844), H02, H03, H04, H05, H06, H07, H09, H10 and H12 haplotypes hold *AVR-Pib* because these isolates were avirulent to the *Pib*-containing monogenic line IRBLb-B (Table 3). The isolate of H08 defeated the resistance of *Pib* because this isolate was virulent to the *Pib*-containing monogenic line IRBLb-B (Table 3). Furthermore, Pot2 and Pot3 inserted in the 5′ UTR of the *Pib* gene were identified in six isolates and 30 isolates (Table 3; Figure 1), respectively, and these isolates were virulent to the *Pib*-containing monogenic line IRBLb-B (Table 3). These findings suggested that the insertion of TEs (Pot2 and Pot3) and small segments of the nucleotide in the promoter region, and the nuclear substitution in the ORF region, resulted in a variation of *AVR-Pib* from avirulence to virulence, and that the diverse mutations of the *AVR-Pib* allele of *M. oryzae* were involved.

### 2.3. Distribution of Haplotypes of AVR-Pib of M. oryzae

Among 12 *AVR-Pib* haplotypes, none were identical to the original *AVR-Pib* (GenBank accession number, KM887844) (Table 2). Eight haplotypes, as well as the Pot2 and Pot3 insertion, were detected in 50 *M. oryzae* isolates from western Yunnan Province (Table 4). Five haplotypes, as well as Pot3 and Pot3 reverse-insertion, were identified in 23 *M. oryzae* isolates from central Yunnan Province. Three haplotypes, as well as Pot2 and Pot3 reverse-insertion, were identified in 15 isolates of *M. oryzae* from northeastern Yunnan Province. Three haplotypes and Pot2 insertion were detected in 19 isolates from northeastern Yunnan Province. Three haplotypes and Pot3 insertion were identified in six *M. oryzae* isolates from southeastern Yunnan Province. Three haplotypes were identified in 13 isolates of *M. oryzae* from northwestern Yunnan Province (Table 4). Eleven and nine haplotypes were detected in *GJ* rice- and *XI* rice production areas, and the diversity index (DI) of haplotypes was 0.84 and 0.79 for these areas, respectively. The DI of *AVR-Pib* was 0.72, 0.71, 0.70, 0.63, 0.59 and 0.54 for southeastern, central, western, southwestern, northeastern and northwestern Yunnan Province, respectively (Table 4). 

In brief, the DI of *AVR-Pib* alleles in Yunnan Province was in the order: southeastern > central > western > southwestern > northeastern > northwestern. The DI of *AVR-Pib* alleles in the *GJ*-rice production area was higher than that in the *XI* rice production area. These results indicate that the genetic divergence of *AVR-Pib* of *M. oryzae* in each rice-growing region occurred depending on each field’s condition.

Eighteen nucleotide variable sites in *AVR-Pib* alleles were identified (Table 2; Appendix A). A haplotype network based on sequence variations of 90 isolates of L1 alleles was developed (Figure 2). Four main lineage branches (A to D) of *AVR-Pib* were divided among 90 field isolates (Figure 2), and a different evolution of *AVR-Pib* among them was noted. Isolates of B and D lineage branches of *AVR-Pib* were avirulent to IRBLb-B (with *Pib*) (Figure 2; Table 3). Isolates of H11 of the A-evolved branch and H08 of the C-evolved branch were virulent to the rice-blast-resistant gene *Pib*, respectively (Figure 2; Table 3). These data suggested that the A and C branches of *AVR-Pib* had evolved to virulence from avirulent origins *via* base substitution and insertion, and evaded the recognition of the rice-blast-resistance gene *Pib* in field isolates. The virulence of H08 and H11 was identified in southeastern and western Yunnan Province (Table 4). Moreover, TE insertion in rice samples in all regions except northwestern Yunnan Province (Table 4) suggested that virulent evolution of *AVR-Pib* occurred in most rice production areas of Yunnan Province.

### 2.4. Selection Pressure on AVR-Pib in M. oryzae

The natural-selection pressure on *AVR-Pib* was calculated by Tajima’s neutrality test on 126 *AVR-Pib* CDS sequences: the Tajima’s *D* value was not significantly different from zero (*D* = −1.61687; 0.10 > P > 0.05) (Appendix A). This result suggested that *AVR-Pib* may suffer neutral selection and evolve neutrally in the population of *M. oryzae*. Furthermore, the results of three positive-selection models kept a higher similarity (Appendix A). The “sliding window” under M8, M8a and M7 models showed values of Ka/Ks (Ka, rate of nonsynonymous substitutions; Ks, rate of synonymous substitutions) across all 74 amino acids (Appendix A). The Ka/Ks value of all sites was >1 under the M8 and M8a model, and the value was 1 under the M7 model for entire residues. These results implied that the sites may have suffered from neutral selection. These findings suggest that the *AVR-Pib* maybe under a neutral evolution.

### 2.5. Adaption of TE Insertion in AVR-Pib

We wished to confirm the host (rice and non-rice) selection pressure on TE insertion in *AVR-Pib*. A total of 27 isolates from *O. rufipogon* (with *Pib* homologs) [16], *Digitaria sanguinalis*, *Eleusine indica*, *E. coracana* and *Musa nana* Lour, which were stored in our lab, and 5 isolates of the genome sequence from *Lolium perenne* Linn (2 isolates), *Setaria viridis* (Linn.) Beauv. (1 isolate) and *Triticum aestivum* Linn (2 isolates) from Genbank were selected and analyzed (Appendix A). The *AVR-Pib* allele was not detected in the isolate from *D. sanguinalis*, *M. nana* Lour, *Lolium perenne* Linn and *Setaria viridis* (Linn.) Beauv. (Appendix A). Only the L1 genotypes (with the expected size) of *AVR-Pib* were detected in isolates from *E. indica*, *E. coracana* and *Triticum aestivum* Linn, suggesting that these isolates did not have a TE insertion in the *AVR-Pib* allele. Three genotypes of L1, L2 (with TE insertion) and L3 (with TE insertion) of *AVR-Pib* were detected in 18 isolates from the *Pib* homolog-containing *O. rufipogon*, and the isolate of YN441 (with the H9 haplotype of *AVR-Pib* and identical with the original haplotype of KM887844) was virulent to *O. rufipogon* (Figure 3). These findings showed that the diversification of *AVR-Pib* of *M. oryzae* was dependent upon the *Pib* homolog-containing *O. rufipogon*, and that the variation in TE insertion in *AVR-Pib* could be selected and adapted to rice and other Gramineae species. 

### 2.6. Phylogeny of Pib Allele Partial to CDS Regions

Fifty-seven sequences of *Pib* were obtained from GenBank (Appendix A). Eleven of them were from five wild-rice species (seven from *O. rufipogon*, one from *O. meyeriana*, one from *O. officinalis*, one from *O. longistaminata* and one from *O. nivara*), and 46 accessions from *O. sativa*, including the original *Pib* (GenBank accession number, AB013448.1) (Appendix A). These sequences were aligned. A minimum-evolution phylogenetic tree was constructed based on the nucleotide sequences of exon 1 of *Pib* in 34 accessions and partial regions of exon-3 nucleotide sequences (from 7633 to 8484 of AB013448.1) of *Pib* in 49 accessions, respectively (Figure 4). Exon 1 of *Pib* in wild-rice species (*O. rufipogon*, *O. meyeriana*, *O. officinalis* and *O. longistaminata*) was close to that in *O. sativa* (Figure 4B). The DQ317978.1 group of the wild rice *O. rufipogon* shared >90% identity with the nucleotide sequences of the JN564624.1 group of *Indica*. Two major clades emerged in one part of exon 3 of *Pib* (Figure 4C). One clade contained two wild-rice species (*O. rufipogon* and *O. nivara*) and *O. sativa*. The EF642422.1 group of the wild-rice species *O. nivara* shared >90% identity in nucleotide sequences with the EF642423.1 group of *O. sativa*. The EF642442.1 group of *O. rufipogon* shared >75% identity of nucleotide sequences with the EF642423.1 group of *O. sativa*. The other clade contained *O. rufipogon* and *O. sativa*. The EF642440.1 group of the wild-rice species *O. rufipogon* shared >75% identity of nucleotide sequences with the EF642433.1 group of *O. sativa*. The isolate of YN441 (with the H9 haplotype of *AVR-Pib*) was virulent to *O. rufipogon*, *O. meyeriana*, and *O. officinalis* (Figure 3). These results suggested that different regions of the *Pib* gene may have suffered different selection pressures in the host rather than domestication.

## 3. Discussion

We identified 12 new haplotypes, as well as Pot2 and Pot3 insertion in the *AVR-Pib* DNA sequences among rice blast isolates from different rice-growing areas in Yunnan Province. The many virulent isolates to *Pib*-containing rice varieties implied that *Pib* was overcome in these rice-growing regions because of the massive exploitation of *Pib* in China. 

*Pib* alleles have been used widely and have shown strong resistance to disease in China [14]. Complete deletions have occurred in *AVR-Pib* sequences among field isolates of *M. oryzae* from various rice-growing regions of Guangdong, Hunan and Liaoning Provinces [18]. Moreover, TE insertion has occurred in *AVR-Pib* in *M. oryzae* isolates from south and northeast China [18] and the Philippines [17]. These data are consistent with our results. The L1 genotype of *AVR-Pib* identified in rice blast isolates collected from rice fields implied that *Pib* has been effective in preventing rice blast. Li and colleagues showed that rice cultivars with *Pib* were resistant to 74.9% of isolates (282 isolates) from Yunnan Province [8]. The corresponding value was 2.1% in Guangdong Province (146 isolates were tested) [28], and the percentage resistance was <31% in Hunan Province [29]. These results show that *Pib* alleles had poor effects in these rice production areas. The further inspection of variation in *AVR-Pib* DNA sequences in these isolates could reveal the molecular evolutionary patterns of *AVR-Pib* and predict the durability and effectiveness of *Pib* allele-mediated resistance under field conditions in rice production regions.

*AVR-Pia*, *AVR-Pii* and *AVR-Pita1* located on telomere regions tend to be unstable, and effective mutants in these genes were identified [21,30,31]. The retrotransposon (MINE) insertion in the *ACE1* gene [23] and Pot3 insertion in *AVR-Pita1* [32,33] and *AVR-Pizt* [22] caused new virulent alleles. TE (Pot2 and Pot3) insertion, complete absence, segmental deletion and a point mutation were found in *AVR-Pib* alleles, all of which lead to a gain of virulence [18]. Three expression patterns were identified among different haplotypes of *AVR-Pib* [18]. Recently, insertion of a Pot3 transposon in *AVR-Pib* was shown to mediate the loss of function of *AVR-Pib* in all 248 isolates collected from the Philippines [17]. These findings showed that rice blasts can use transposons to suppress the expression of *AVR* genes to defeat the rice blast resistance gene. The *AVR-Pib* allele was identified in nearly half of rice blast isolates (44.3%) in Yunnan Province (Table 1). This percentage was higher than that in rice blast isolates in Jilin and Heilongjiang Provinces, but lower than those in Guangdong, Hainan and Liaoning Provinces [18]. Meanwhile, 28.6% of isolates contained a TE insertion in *AVR-Pib*. Among them, 21.4% isolates contained a Pot3-element insertion and 2.4% of isolates contained a reverse Pot3-element insertion, and 4.8% isolates contained a Pot2-element insertion in *AVR-Pib* of *M. oryze* from Yunnan Province (Table 4; Figure 1). These insertions resulted in the variation from avirulence to virulence to the corresponding *R* gene. Several nucleotide variations in *AVR-Pib* alleles were identified, which led to variations in amino acids and implied that *AVR-Pib* alleles suffer from strong selection pressure in rice production regions of Yunnan Province. Whereas, mutations that impact gene expression are further acknowledged by transcript level.

We observed no TE insertions in *AVR-Pib* of isolates from *E. indica* or *E. coracana,* and TE insertion in *AVR-Pib* was selected by the host, data that are consistent with the results of Zhang et al. [18]. Pot2 and Pot3 insertions were identified in the *XI*-rice-growing areas, whereas only Pot3 insertion was identified in *GJ*-rice-growing areas of Yunnan Province. TE insertion of *AVR-Pib* was noted in all rice-growing regions except northwestern Yunnan Province, and Pot3 insertion was distributed mainly in western Yunnan Province. These results showed that the virulent *AVR-Pib* alleles were involved in most of the rice-growing regions of Yunnan Province. Hence, the monitoring of these virulent alleles in field populations is important for employing *Pib*-containing rice varieties.

Various mutations were identified in CDS regions of *AVR-Pib*, and 12 *AVR-Pib* haplotypes were found based on 18 variant nucleotides among 90 isolates of L1 alleles collected from Yunnan Province (Table 2). Six new variant amino acids of the *AVR-Pib* loci variants were identified in the 90 *M. oryze* isolates in the present study, and resulted in the identification of four novel haplotypes. A more holonomic network was constructed based on the new variations among different alleles of *AVR-Pib*. The putative and secreted proteins of AVR-Pib in 126 isolates were identified (Table 3), and they were in accordance with the results of Zhang et al. [18]. Nine isolates had variations at the amino acid position F54L; three isolates had variations at the amino acid positions of E46V, Y53S and F54V; one isolate had variations at the amino acid positions of F47L, I49T and R50G (Table 3). These isolates were virulent to the monogenic line IRBLb-B (with *Pib*), suggesting that these amino acids are crucial for avirulent function. Isolates of H11 with insertion of ATTA in the 5′ UTR may change *AVR-Pib* expression and cause a loss of the avirulent function (Table 2 and Table 3; Appendix A).

In the course of interactions and co-evolution between pathogens and plants, the *R* genes of plants can discern the cognate *AVR* genes of pathogens and inspire immunity [1]. The genetic variation of the *AVR* genes of the pathogen is dependent upon the *R* genes of the host and changeable environmental conditions. The DI of *AVR-Pib* was higher in *GJ* rice production areas than that of *XI* rice production areas (Table 4). Different variations were observed in *AVR-Pib* between *XI* rice- and *GJ* rice production areas (Table 4). These findings imply that the adaptive mutations of *AVR-Pib* occurred in Yunnan Province under natural conditions, and these results were similar in previous studies [18]. 

Yunnan Province is abundant in genetic resources of rice. The wild species of *O. officinalis*, *O. meyeriana* and *O. rufipogon* also coexist in this province [34]. More than 5000 rice accessions germplasms have been conserved in Yunnan Province. Among them, 12 out of 227 accessions carried the *Pib* resistance gene screened by the resistance gene identification using different isolates [34]. *Pib* gene homologs were identified in wild rice *O. rufipogon* from Yuanjiang County [16], and four genotypes (L0 to L3) of *AVR-Pib* were detected in *M. oryzae* and *O. rufipogon* in Yuanjiang County. TE insertion of L2 and L3 genotypes of *AVR-Pib* was absent in the isolates from *D. sanguinalis* and *M. nana* Lour. These results suggest that the adaptive variation of *AVR-Pib* is involved during interactions and co-evolution between *AVR-Pib* of *M. oryzae* and *Pib* of *O. rufipogon*. The Tajima’s *D* value of −1.61687 (Appendix A) indicates that *AVR-Pib* loci may suffer from a purifying selection by the corresponding *R* gene in Yunnan rice production areas. 

Massive variations and stepwise mutations in *AVR-Pib* of rice blast isolates were observed in Yunnan Province (Table 2; Figure 2), which suggests that there is an abundant diversity of rice accessions and *M. oryzae* isolates in Yunnan Province. Pot2 insertion in *AVR-Pib* was found in western, southwestern and northeastern Yunnan Province. Pot3-reversed insertion was found in central and northeastern Yunnan Province, which was not observed in the previous studies [18]. Moreover, Pot3 insertion occurred mainly in western Yunnan Province. Pot3 insertion of *AVR-Pib* was found in *GJ* rice and *XI* rice production areas, whereas Pot2- and Pot3-reversed insertion was found only in the *XI* rice production area and *GJ* rice production area, respectively (Table 4). The virulent haplotype of H11 was detected in the *XI* rice production area and *GJ* rice production area, and H08 was detected in the *XI* rice production area. These data showed a high variation of *AVR-Pib* in different rice-growing regions, which may be due to rice variety and the environment.

The stepwise mutations that result in a loss of avirulence function have been identified in *AVRL567* [35] and *AVR-Pik* [36,37,38,39]. Based on the *Pib* homologs identified by Yang et al. [16], and our result for *AVR-Pib* in the present study, the potential interactions and co-evolution of *AVR-Pib* alleles in *M. oryzae* and *Pib* alleles of rice were constructed (Appendix A). The *AVR-Pib* homolog L1 (H01) originated from an ancestral *M. oryzae* gene. The *Pib* allele (87-bp deletion in exon 1 of *Pib*) in *O*. *rufipogon* could not recognize the L1 alleles of *AVR-Pib*. Thus, the other *Pib* allele (gained 87 bp in exon 1 of *Pib*) in cultivated rice evolved to recognize the L1 alleles (H01) of *AVR-Pib*, whereas the altered alleles L2 and L3 evolved to virulence from avirulent origins by TE insertion, base substitution (H08) and segment insertion (H11) to avoid recognition by *Pib* (Table 2; Appendix A). These actions indicated a stepwise evolution of *AVR-Pib* as well as *Pib* interaction and co-evolution. Intriguingly, the *AVR-Pib* alleles H08 and H11 were derived from H01, and could escape recognition by *Pib* (Table 2; Appendix A), but several extinct or missing haplotypes were not identified in the sample (Figure 2). These findings imply that: (i) the *AVR-Pib* loci of *M. oryzae* evolved gradually during the interaction and coevolution between the *Pib* loci of *M. oryzae* in field conditions; (ii) the genome organization of the *AVR-Pib* locus is much more intricate than anticipated.

## 4. Materials and Methods

### 4.1. Blast Isolates, Rice Accessions, Culture and Pathogenicity Identification

The rice blast fungus single spores were obtained from infected leaves or panicles incubated on moist filter paper in a Petri dish at room temperature for 24 h according to Jia et al. [38], and the tests made fulfilled the Kock’s postulates for all isolates. The seedlings of the rice monogenic line IRBLb-B (which contains *Pib*) and a susceptible cultivar Lijiangxintuanheigu (LTH; does not contain *Pib*) were used for pathogenicity assays (the seeds were acquired originally from Cailin Lei). Four thousand isolates were collected from six rice-growing regions from 1997 to 2012 in Yunnan Province, and a total of 366 isolates are selected from six rice-growing regions as representative isolates. The total of 366 isolates of *M. oryzae* in the present study were the same in the published paper [38]. The storing and culturing methods of the isolates were described in the published paper [38]. Disease reactions were referred to as the method of Jia et al. [39]. In a few words, when rice seedlings were in the 3- to 4-leaf stage, they were inoculated with a spore suspension (1–5 × 10^5^ spores/mL containing 0.05% Tween 20). After innoculation, rice seedlings were put into a plastic bag and sealed securely to keep a high relative humidity of 90–100% at 25 °C for 24 h in the dark. Then, the plants were shifted into a greenhouse for 6 days to develop the disease lesion extension fully. 

Disease reactions were monitored externally on the second youngest leaf based on the number and degree of lesions using a 0-to-5 disease scale (Appendix A). The disease scale method was described in the published paper [38]. Five seedlings at a time, repeating twice, were arranged in the experiment. In addition, the average value of disease scales was used to discriminate resistance versus susceptibility. The disease reaction of wild rice species *O*. *rufipogon*, *O. meyeriana* and *O. officinalis* (which were conserved in Yunnan Academy of Agricultural Sciences) was determined using a detached-leaf method, as described by Jia et al. [39]. One virulent blast isolate was used for inoculation. Disease reactions were evaluated 5–7 days after inoculation.

### 4.2. DNA Extraction, PCR Amplification and DNA Sequencing

A culturing method of vegetative mycelia of *M. oryzae* isolates was described in the published paper [38]. The genomic DNA of each isolate was extracted from vegetative mycelia by CTAB method [40]. The primers AvrPibF1 (5′-GGACAAGGGAGGCAAATCTAAC-3′) and AvrPibR1 (5′-ATGCCGACAATGCGAGGTAT-3′) were used to amplify the *AVR-Pib* allele, as well as for sequencing according to the method of Zhang et al. [18]. Each PCR reaction was amplified in a total reaction volume of 50 µL containing the following components: 25 µL of 2× Taq PCR MasterMix (Tiangen Biotech, Beijing, China), 1 µL (10 µM) of each primer, 2 µL of template DNA, and 21 µL of ddH_2_O (provided in the Tiangen kit). PCR procedure was conducted in a C1000 Touch™ thermal cycler (Bio-Rad Laboratories, Hercules, CA, USA) in the following steps: initial denaturation at 94 °C for 3 min, followed by 29 cycles at 94 °C for 45 s, 55 °C for 45 s and 72 °C for 2.5 min, and a final extension at 72 °C for 5 min. Each reaction was repeated twice. The size of the PCR products was valued by a DNA marker (DL2000, Tiangen Biotech). Amplicons were sequenced twice by Life Technologies Biotechnology (Shanghai, China).

### 4.3. Data Analyses

DNA sequences of *AVR-Pib* were assembled and aligned using DNASTAR v7.1.0 (www.dnastar.com/, 15 January 2019). DnaSP v5.10.01 [41] was used for the calculation of polymorphic sites (*π*), the number of DNA haplotypes and the sliding window. TCS1.21 (http://darwin.uvigo.es/, accessed on 1 February 2019) [42] was used for analyses of the haplotype network of *AVR-Pib*. The DI (haplotype diversity index) was counted in *M. oryzae* populations following the method of Fontaine et al. [43]:

DI = (1 − ∑^n^_i = 1_p_i_^2^)

where p_i_ is the frequency of haplotype i in a population. Tajima’s neutrality test was conducted using MEGA X (www.megasoftware.net, accessed on 12 February 2019) [28]. The Selection Server program (http://selecton.tau.ac.il, accessed on 19 February 2019) was used for analyses of purifying the selection. The purifying of the selected sites of AVR-Pib was identified by used three models: M8 (positive selection enabled, beta + w ≥ 1), M7 (beta, null model) and M8a (beta + w = 1, null model). Then, the sliding window of purifying the selected sites of AVR-Pib was drawn under the M8, M7 and M8a models by Excel™ (Microsoft, Redmond, WA, USA). MEGA X [28] was used for the construction of phylogenetic trees by the minimum evolution method [44]. The SWISS-MODEL (http://swissmodel.expasy.org, accessed on 18 February 2019) with ProMod v3.7.0 was used to build the protein homology model. The significant difference of distribution of *AVR-Pib* alleles and avirulence isolates of *M. oryzae* in each region was analyzed by Excel software with CHITEST.

## 5. Conclusions

We detected twelve novel haplotypes in the field population by using 90 isolates and a transposable element (TE) insertion in 36 of 126 isolates, constructing a complex network of *AVR-Pib* alleles and assessing the efficacy of *Pib* alleles in rice production areas of Yunnan; we also analyzed the adaption of TE insertion of *AVR-Pib* in the isolates from a different host. Our findings support the hypothesis that functional *AVR-Pib* possesses varied sequence structures and can escape surveillance by hosts via multiple variation manners. Haplotype H08 and H11 can overcome all detected *Pib* alleles to date, and Pot insertion can change the avirulent function of *AVR-Pib*. Despite the H08, H11 haplotypes and TEs insertions have low frequencies; the monitoring of these alleles in field populations is critical because of their high risk for *Pib*-holding rice varieties. The TE insertion was not detected in the *AVR-Pib* allele in the isolates from *E. indica*, *E. coracana*,and *Triticum aestivum* Linn, while three genotypes of *AVR-Pib* were detected in isolates from *O. rufipogon*. The selected and adapted variation of TE insertion in *AVR-Pib* is the occurrence of the long-term co-evolution between *M. oryzae* and hosts (rice and other Gramineae species).

## Figures and Tables

**Figure 1 ijms-24-15542-f001:**
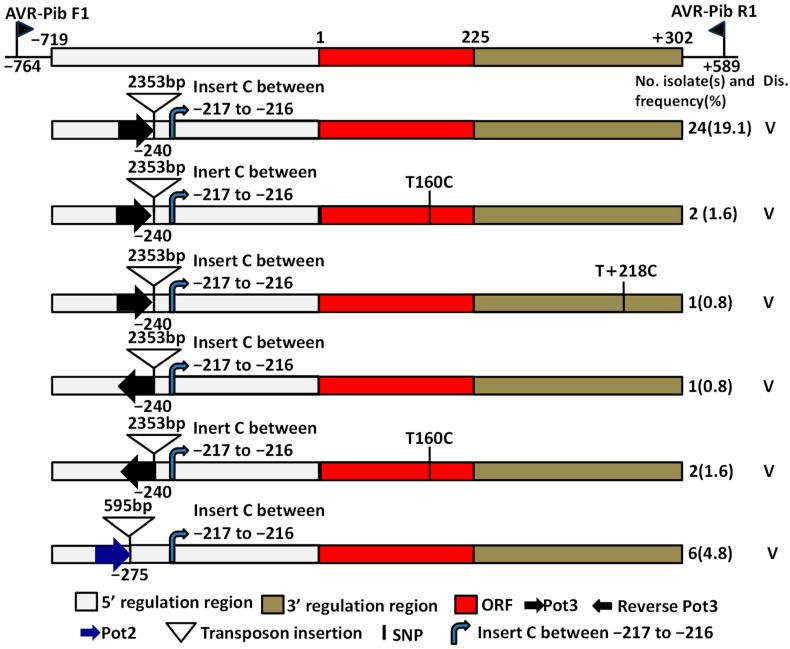
Characterization of allelic variation at *AVR-Pib*. The functional nucleotide polymorphic maps of the six natural alleles in 126 isolates of *M. oryzae* in Yunnan Province. Dis. indicates disease reaction on monogenic line IRBLb-B (containing *Pib*), V indicates the isolates were virulent to IRBLb-B.

**Figure 2 ijms-24-15542-f002:**
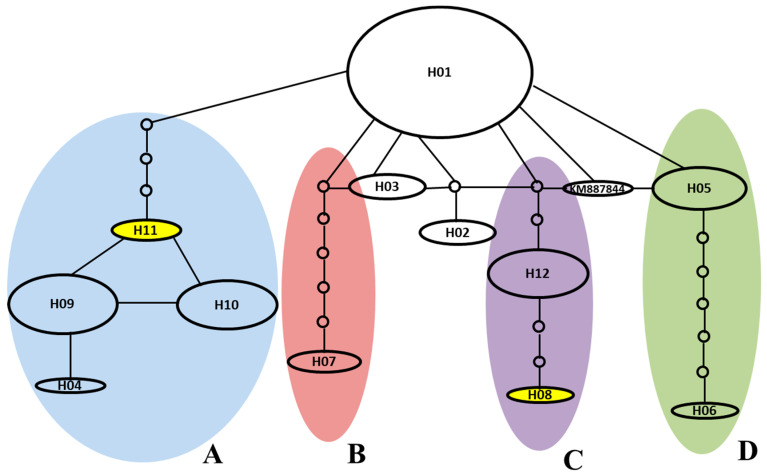
The haplotype network for the 12 *AVR-Pib* alleles. Haplotype network analysis was performed using TCS1.21 (http://darwin.uvigo.es/, accessed on 12 March 2019.). The original *AVR-Pib* allele was designated as the H01 haplotype in the network. Each haplotype was separated by mutational events. The node in the network represents an extinct or a missing haplotype not found among the samples. Each haplotype was separated by mutational events. All haplotypes were displayed as circles. The size of the circles corresponds to the haplotype frequency. The KM887844 (GenBank Accession No.) of *AVR-Pib* was obtained from GenBank. White color indicates avirulent to the *Pib* gene and yellow color indicates virulent to the *Pib* gene. A to D, four major haplotypes of *AVR-Pib* in Yunnan Province of China, are shaded.

**Figure 3 ijms-24-15542-f003:**
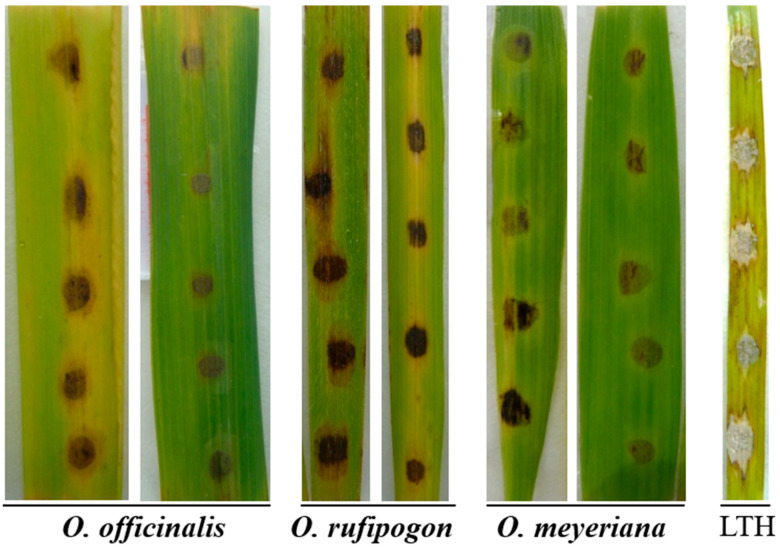
Disease reaction of the identification isolate of YN441 (with the H9 haplotype of *AVR-Pib* which was identical with the original haplotype of KM887844) on *Oryza officinalis*, *O. rufipogon* and *O. meyeriana*. LTH: LijiangxinTuanHeigu.

**Figure 4 ijms-24-15542-f004:**
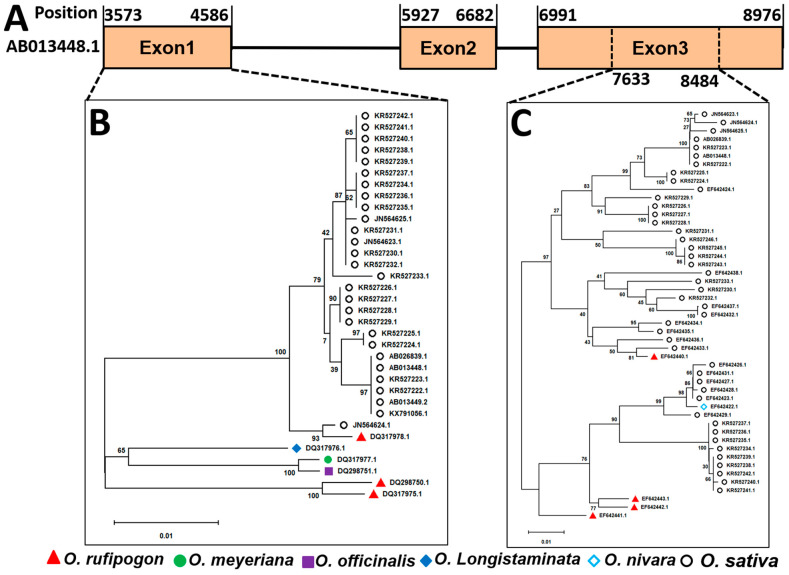
Phylogenetic tree constructed with the nucleotide sequences of *Pib* gene and different partial CDS regions from wild rice and *O. sativa* using minimum evolution method of MEGA X. The numbers associated with individual branches indicate confidence levels based on 1000 bootstrap replicates. (**A**)**,** structure of *Pib* from AB013448.1 (GenBank ID); (**B**)**,** the phylogenetic tree constructed based on the nucleotide sequences of exon1 of *Pib* regions from 34 accessions. (**C**)**,** the phylogenetic tree constructed based on the nucleotide sequences of partial exon3 (from 7633 to 8484 of AB013448.1) of *Pib* from 49 accessions. All accessions of *Pib* were obtained from GenBank.

**Table 1 ijms-24-15542-t001:** Frequency of *AVR-Pib* genotypes and avirulent isolates of *Magnaporthe oryzae* collected from Yunnan, China to IRBLb-B.

Locations	No. of Isolates	PCR Detection ^a^				Pathogenicity Assay ^b^
		Genotype and No. of Isolates and Frequency (%)				No. of Avirulence Isolates and Frequency (%)
		L1	L2	L3	Total Isolates and Frequency (%)	
Central	54	22 (40.7)	3 (5.6))	0	25 (46.3) ^B^	41 (75.9) ^AB^
Northeastern	72	23 (31.9)	3 (4.2)	0	26 (36.1) ^B^	54 (75.0) ^B^
Northwestern	15	11 (73.3)	0	0	11 (73.3) ^A^	15 (100) ^A^
Southeastern	33	10 (30.3)	6 (18.2)	1 (3.0)	17 (51.5) ^B^	19 (57.6) ^C^
Southwestern	28	9 (32.1)	8 (28.6）	2 (7.1)	19 (67.9) ^A^	15 (53.6) ^C^
Western	164	29 (17.7)	33 (20.1)	2 (1.2)	64 (39.0) ^B^	79 (48.2) ^C^
Total	366	104 (28.4)	53 (14.5)	5 (1.4)	162 (44.3)	223 (60.9)
*XI*	149	31 (20.8)	34 (22.8)	5 (3.4)	70 (47.0) *	69 (46.3) **
*GJ*	217	73 (33.6)	19 (8.8)	0	92 (42.4) *	154 (71.0) **
Total	366	104 (28.4)	53 (14.5)	5 (1.4)	162 (44.3)	223 (60.9)

^a^ L1 indicates the *AVR-Pib* genotype with the expected size (1231 bp), L2 and L3 indicates the *AVR-Pib* genotype with TE insertion (L2 with 3100 bp and L3 with both 1231 bp and 3100 bp). The frequencies in bracket. The superscript of A and B indicates the significant difference at 0.01 level, and * indicates non significant. ^b^ Indicates pathogenicity assay on monogenic line IRBLb-B containing *Pib*. *XI* and *GJ* indicates *Xian*/*Indica* and *Geng*/*Japonic*, respectively. The frequencies in bracket. The superscript of A, B and C indicates the significant difference at 0.01 level; ** indicates significant difference at 0.01 level.

**Table 2 ijms-24-15542-t002:** Haplotypes of *AVR-Pib* loci in *Magnaporthe oryza*e field populations of Yunnan, China.

Haplotype	No. of Isolates	% of Total	Variant Locus ^a^																	
			5′ UTR								CDS Regions						3′ UTR			
			−338	Between −325 and −326	Between −239 and −240	Between −216 and −217	−192	−175	Between −210 and −211	−93	137	141	146	148	158	160	+70	+154	+218	+232
KM887844			T	-	-	-	C	C	-	T	A	T	T	C	A	T	C	G	A	C
H01	33	26.2	.	.	.	C	.	.	.	.	.	.	.	.	.	.	.	.	.	.
H02	4	3.2	.	.	.	C	.	.	.	A	.	.	.	.	.	.	.	T	.	.
H03	4	3.2	.	.	.	C	T	.	.	.	.	.	.	.	.	.	.	.	.	.
H04	1	0.8	.	ACTTA	.	C	.	.	.	.	.	.	.	.	.	.	T	.	.	.
H05	8	6.3	C	.	.	C	.	.	.	.	.	.	.	.	.	C	.	.	.	.
H06	1	0.8	C	.	ACGTTA	C	.	.	.	.	.	.	.	.	.	C	.	.	.	.
H07	3	2.4	.	.	.	C	.	T	.	.	T	.	.	.	C	G	.	.	.	A
H08	1	0.8	.	.	.	C	.	.	ACA	.	.	A	C	G	.	.	.	.	.	.
H09	13	10.3	.	ACTTA	.	C	.	.	.	.	.	.	.	.	.	.	.	.	.	.
H10	10	7.9	.	AGTTA	.	C	.	.	.	.	.	.	.	.	.	.	.	.	.	.
H11	2	1.6	.	ATTA	.	C	.	.	.	.	.	.	.	.	.	.	.	.	.	.
H12	10	7.9	.	.	.	C	.	.	ACA	.	.	.	.	.	.	.	.	.	.	.
Pot2	6	4.8	−275 insert Pot2		.	C	.	.	.	.	.	.	.	.	.	.	.	.	.	.
Pot3 rev-A	1	0.8	−240 insert Pot3		.	C	.	.	.	.	.	.	.	.	.	.	.	.	.	.
Pot3 rev-B	2	1.6	−240 insert Pot3		.	C	.	.	.	.	.	.	.	.	.	C	.	.	.	.
Pot3-A	24	19.1	−240 insert Pot3		.	C	.	.	.	.	.	.	.	.	.	.	.	.	.	.
Pot3-B	2	1.6	−240 insert Pot3		.	C	.	.	.	.	.	.	.	.	.	C	.	.	.	.
Pot3-C	1	0.8	−240 insert Pot3		.	C	.	.	.	.	.	.	.	.	.	.	.	.	C	.

^a^ Indicates the same with KM887844 (GenBank Accession No.). The KM887844 of *AVR-Pib* was obtained from GenBank. rev: indicates reverse insertion of Pot3 in *AVR-Pib*.

**Table 3 ijms-24-15542-t003:** Variation of the *AVR-Pib* loci proteins in *M. oryza*e populations of Yunnan, China.

Haplotype	Variant Locus ^a^						Disease Reaction ^b^
	46	47	49	50	53	54	
KM887844	E	F	I	R	Y	F	
H01	.	.	.	.	.	.	24R + 5M + 4?
H02	.	.	.	.	.	.	3R + 1M
H03	.	.	.	.	.	.	4R
H04	.	.	.	.	.	.	1R
H05	.	.	.	.	.	L	7R + 1?
H06	.	.	.	.	.	L	1R
H07	V	.	.	.	S	V	3R
H08	.	L	T	G	.	.	1S
H09	.	.	.	.	.	.	11R + 2M
H10	.	.	.	.	.	.	9R + 1M
H11	.	.	.	.	.	.	2S
H12	.	.	.	.	.	.	9R + 1M
Pot2	.	.	.	.	.	.	7S
Pot3 rev-A ^c^	.	.	.	.	.	.	1S
Pot3 rev-B	.	.	.	.	.	L	2S
Pot3-A	.	.	.	.	.	.	22S + 2M
Pot3-B	.	.	.	.	.	.	2S
Pot3-C	.	.	.	.	.	L	1S

^a^ Indicates the same with KM887844. ^b^ Indicates pathogenicity assay on the monogenic lines IRBLb-B containing the resistant gene of *Pib*. R, M and S indicate disease reaction were resistant, moderate resistant and susceptible, respectively; Ex. 24R indicated 24 isolates were avirulent to IRBLb-B; and ? indicates unknown. ^c^ rev: indicates reverse insertion of Pot3 in *AVR-Pib*.

**Table 4 ijms-24-15542-t004:** Haplotype distribution of *AVR-Pib* in different Yunnan rice-growing regions.

Haplotype	Regions						Production ^c^	
	Central	Northeastern	Northwestern	Southwestern	Southeastern	Western	*XI*	*GJ*
H01	11(47.8) ^a^	9(60.0)	2(15.4)	10(52.6)	0	1(2.0)	10(20.8)	23(29.5)
H02	0	0	0	4(21.1)	0	0	4(8.3)	0
H03	4(17.4)	0	0	0	0	0	0	4(5.1)
H04	0	0	0	0	0	1(2.0)	0	1(1.3)
H05	1(4.3)	1(6.7)	3(23.1)	0	2(33.3)	1(2.0)	2(4.2)	6(7.7)
H06	0	0	0	0	1(16.7)	0	1(2.1)	0
H07	0	0	0	0	0	3(6.0)	0	3(3.8)
H08	0	0	0	0	0	1(2.0)	1(2.1)	0
H09	2(8.7)	0	0	0	0	11(22.0)	0	13(16.7)
H10	2(8.7)	0	8(61.5)	0	0	0	0	10(12.8)
H11	0	0	0	0	1(16.7)	1(2.0)	1(2.1)	1(1.3)
H12	0	3(20.0)	0	1(5.3)	0	6(12.0)	6(12.5)	4(5.1)
Pot2	0	1(6.7)	0	4(21.1)	0	1(2.0)	6(12.5)	0
Pot3 rev	2(8.7)	1(6.7)	0	0	0	0	0	3(3.8)
Pot3	1(4.3)	0	0	0	2(33.3)	24(48.0)	17	10(12.8)
Total	23	15	13	19	6	50	48	78
No. of haplotypes	7	5	3	4	4	10	9	11
Index of diversity ^b^	0.71	0.59	0.54	0.63	0.72	0.70	0.79	0.84

^a^ Number and frequency (in bracket) of isolates of each haplotype. ^b^ Diversity index was calculated as the frequency of haplotypes types in the *M. oryzae* population following Fontaine’s method [28]. Diversity index = (1 − ∑^n^_i=1_p_i_^2^) (where pi is the frequency of the haplotype i in a population). ^c^ *XI* and *GJ* indicates *Xian*/*Indica* and *Geng*/*Japonic*, respectively.

## Data Availability

All relevant data are presented within the paper and its Appendix A. The nucleotide sequences of novel *AVR-Pib* alleles from these isolates have been deposited in GenBank (accession numbers: OR361637 to OR361654).

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
