# Peer review of "Insertion of Transposable Elements in AVR-Pib of Magnaporthe oryzae Leading to LOSS of the Avirulent Function"

_ijms, 2023, doi:10.3390/ijms242115542_

Round 1

Reviewer 1 Report

This work is aimed to study insertation of transposable elements in AVR-Pib of Magnaporthe oryzae which lead to loss of the avirulent function. The study desing is acceptable. The results contains some elements that worth to publish however the study need to be revised before considering publication.

Suggestions:

Give a conclusion sentence at the end of the Abstract.

Table 1: Give in full M. oryzae in the title. Give explanation for IRBLb-B in the footnotes.  Why do not you made significance level at 0.05? 'frequency' and not 'Frequency'. Are frequencies in bracket?  

Table 2. Give in full M. oryzae in the title.

L247: Footnote a: delete point before Indicates.

Figure 3: Give in full O. officinalis in the title. LTH is not clear.

Figure 4: The tree is too small to read the text (escpecially the part C)

L497-507 and L526-542: Discussion sections without any citations (i.e. comparison with previous results is missing in these sections).

Manuscript shape is not suit to mdpi format.

Author Response

Dear Reviewer,

Our behalf of the all coauthors I thank your constructive comments. The manuscript was revised with addition data and was revised to address all of the comments for ijms-2583204. Point to point response is listed below for you to verify.

Reviewer 1

Suggestions:

  1. Give a conclusion sentence at the end of the Abstract.

Responds: we have changed.

  1. Table 1: Give in full M. oryzae in the title. Give explanation for IRBLb-B in the footnotes.  Why do not you made significance level at 0.05? 'frequency' and not 'Frequency'. Are frequencies in bracket?  

Responds: we have changed. We have made significance level at 0.05, and it is non significance. The frequencies are in bracket.

  1. Table 2. Give in full M. oryzae in the title.

Responds: we have changed.

  1. L247: Footnote a: delete point before Indicates.

Responds: The point before Indicates is including in table.

  1. Figure 3: Give in full O. officinalis in the title. LTH is not clear.

Responds: we have changed.

  1. Figure 4: The tree is too small to read the text (escpecially the part C)

Responds: we have changed.

  1. L497-507 and L526-542: Discussion sections without any citations (i.e. comparison with previous results is missing in these sections).

Responds: we have changed.

  1. Manuscript shape is not suit to mdpi format.

Responds: we have changed.

Please let us know if further edits will be needed. Otherwise we will be pleased to know that the present version of the manuscript is acceptable with ijms.

Sincerely

Jinbin Li

Reviewer 2 Report

The manuscript “Insertion of transposable elements in AVR-Pib of Magnaporthe oryzae leading to loss of the avirulent function” by Li et al, explores mutations in avirulence (AVR) genes of the Magnaporthe oryzae, investigating their role in rice resistance. The prevalence of AVR-Pib alleles is assessed among M. oryzae isolates from Yunnan Province, China. Notably, some isolates display altered virulence due to transposable element insertions in AVR-Pib. The study identifies diverse AVR-Pib haplotypes, shedding light on their evolution from avirulent to virulent forms, driven by genetic changes such as base substitutions and TE insertions. These findings hold implications for disease management and plant breeding strategies. Overall, the manuscript is well written, and the results are clearly presented. Specific comments as below:

1. The abstract dives straight into technical details without briefly introducing the significance of rice blast as a disease. Authors should add a sentence that provides context about the economic and agricultural importance of rice and the threat posed by the rice blast disease.

2. While the abstract mentions that the study was conducted in China, it might be helpful to briefly mention the significance of Yunnan Province in terms of rice production or its relevance to the study.

3. While the introduction covers various aspects of the topic, authors should reframe the introduction section into a more organized structure. Break it down into subsections or paragraphs to address specific points, such as the coevolution hypotheses, the significance of rice blast, the interaction between R and AVR genes, the diversity of AVRPib, and the objectives of the study.

4. While the authors have mentioned the importance of using multiple resistant genes to control rice blast, it might be useful to add a sentence or two about the challenges and potential consequences of relying solely on resistant genes in addition to other methods of disease control.

5. Towards the end of the introduction, explicitly connect the presented background information to the gaps or questions that your study aims to address. This will help readers understand why your study is necessary and what it contributes to the field.

6. Instead of repeating the same information multiple times, condense your descriptions. For instance, authors mention the percentage of avirulent isolates in different regions several times. Instead, state this once and then elaborate on other relevant details.

7. The information about DI (Diversity Index) for AVR-Pib alleles in different regions is repeated three times. Mention it once and then refer to it in the subsequent paragraphs.

8. Make sure to consistently use the same terminology for referring to genes and alleles. This will help avoid confusion and enhance clarity.

9. Authors should suggest areas for further research based on the gaps or questions that emerged from the study. This could help guide future investigations in this field.

1   Authors should work on improving sentence structure, clarity, and word choice. Sentences can be made clearer and more concise, avoiding redundancy and repetition. Also, ensure that use of abbreviations is consistent and well-defined.

Reviewer 3 Report

I congratulate the authors for their work and manuscript. the AVR-Pib variants leading to virulence or avirulence of M. oryzae on rice were characterized in detail, revealing possible mutational events that gave rise to this variability. Mutations were detected both on the protein coding sequence and the upstream and downstream untranslated regions. Mutational frequencies also helped the authors to identify possible different selection pressures leading to the current allele population in rice fields. Understanding the evolutionary patterns of the AVR-Pib might provide insights on durability and effectiveness of the Pib allele-mediated resistance. I believe the manuscript provides an important contribution to the research field of rice blast and should be interesting to many readers of IJMS. I have no issues with the current version and will recommend acceptance for publication after English review by the editing team.

English is fine overall but can be improved by the editing team.

Reviewer 4 Report

The manuscript “Insertion of transposable elements in AVR-Pib of Magnaporthe oryzae leading to loss of the avirulent function” reports a survey of 366 isolates of M. oryzae from China about their virulence to rice genotypes with or without Pib. This is not a new approach; however, the authors were able to identify new alleles and tried to relate these alleles with the evolution to more virulent forms.

Along the manuscript, the authors presented the various alleles observed per region, without characterizing properly the rice genotypes in each region as well as the presence of another host that could driven the pathogen evolution. To overcome this the authors should better characterize each region (please take into account that references in Chinese are not acceptable). Another important issue is the absence of any explanation for the main focus on Yunnan Province. 

It is also important to note that it is mandatory to ensure that all the isolates fulfill Koch postulates. This fact is not even referred to.

Other minor issues:

In the section “Results” the authors present the number of isolates and also the percentage,  and because of this, they repeat the same results in different ways. This makes the reading more difficult to follow.

The authors made some phylogenetic studies of the R gene Pib, but they should explain why they did not use all gene sequencing of this gene. 

M&M the authors should explain the isolation methodology adopted and the hosts where the pathogens were isolated.

Round 2

Reviewer 4 Report

THE AUTHORS DID NOT ANSWER ANY OF THE RAISED QUESTIONS!

Round 3

Reviewer 4 Report

The testes made for fulfill the Kock’s postulates still absent in the manuscript.

In parallel the description of fungus isolation is not clearly explained although the authors referred a previous published paper, my suggestion is to include a briefly description of the isolation methodology.

 Line 145 – isolatestested space missing
